# Wearable Spine Tracker vs. Video-Based Pose Estimation for Human Activity Recognition

**DOI:** 10.3390/s25123806

**Published:** 2025-06-18

**Authors:** Jonas Walkling, Luca Sander, Arwed Masch, Thomas M. Deserno

**Affiliations:** Peter L. Reichertz Institute for Medical Informatics of TU Braunschweig and Hannover Medical School, 38100 Braunschweig, Germany; j.walkling@tu-bs.de (J.W.); luca.sander@tu-bs.de (L.S.); arwed.masch@tu-bs.de (A.M.)

**Keywords:** human activity recognition (HAR), activity of daily living (ADL), inertial measurement unit (IMU), FlexTail, body-worn sensors, environment-integrated sensors, wearable sensors, pose estimation, real-time monitoring, time series classification, machine learning

## Abstract

This paper presents a comparative study for detecting the activities of daily living (ADLs) using two distinct sensor systems: the FlexTail wearable spine tracker and a camera-based pose estimation model. We developed a protocol to simultaneously record data with both systems and capture eleven activities from general movement, household, and food handling. We tested a comprehensive selection of state-of-the-art time series classification algorithms. Both systems achieved high classification performance, with average $F_{1}$ scores of 0.90 for both datasets using a 1-second time window and the random dilated shapelet transform (RDST) and QUANT classifier for FlexTail and camera data, respectively. We also explored the impact of hierarchical activity grouping and found that while it improved classification performance in some cases, the benefits were not consistent across all activities. Our findings suggest that both sensor systems recognize ADLs. The FlexTail model performs better for detecting sitting and transitions, like standing up, while the camera-based model is better for activities that involve arm and hand movements.

## 1. Introduction

Clinicians monitor patients in their normal living environment in cases of obesity, depression, and post-surgical aftercare [1]. In contrast to dynamic human activity recognition (HAR), static postures are analyzed, such as sitting, standing, or hinging [2]. According to Natarajan et al., automated (sensor-based) peri-operative patient monitoring is less susceptible to bias than patient-reported outcomes [3]. For instance, the range of motion in activities of daily living (ADL) indicates post-surgical recovery in spine surgery [4,5].

Advances in sensor technology, machine learning, and data processing foster the detection of ADL [6] or HAR, with wearable inertial measurement units (IMUs), ambient sensors, and camera systems [7]. Minh Dang et al. classify HAR them into two broad categories: sensor-based and vision-based approaches [8]. Sensor-based systems utilize wearables, radiofrequency signals, or ambient sensors. Minh Dang et al. further categorize vision-based approaches into RGB imaging, depth imaging, and pose-based classification [8]. However, we prefer distinguishing between body-worn [9] and environment-integrated sensors, where environments such as the private home [10] or vehicle [11] play a major role.

Detecting ADL or HAR is an ongoing field of research. Already in 2009, Marszalek, Laptev and Schmid analyze actions in their context using video cameras [12]. Moreover, Tao et al. use depth cameras to monitor home activity [13], and Sen et al. apply radar [14]. Zhuravchak, Kapshii and Pournaras suggest WiFi channel state information (CSI) and a deep neural network for activity recognition [15]. Atallah et al. apply intertia measurement unitss (IMUs) such as accelerometers for activity recognition [16].

Besides IMUs, researchers and developers have presented smart wearables and applied them for HAR. For instance, Piau et al. presented a smart shoe insole to monitor frail older adults’ walking speed [17].

However, most papers on HAR consider only a single type of input or use sensor fusion [18]. Only a few works directly compare different sensor modalities; for example, Tao et al. [13] compared IMUs with cameras for ADL detection.

In previous work, we evaluated the FlexTail device (MinkTec GmbH, Braunschweig, Germany) that integrates printed electronics into a thin, flexible strip. FlexTail is worn under the clothes and measures bending and torsion of the spine across 18 sensor elements, providing three-dimensional, real-time spinal curvature data without external markers or rigid attachment points [19]. This device is also used to measure range of motion in post-surgical care [20]. If the feasibility of HAR using the FlexTail can be shown, this device would be suitable for providing comprehensive perioperative monitoring of both activity levels and Range-of-Motion (ROM).

In this work, we select an off-the-shelf video camera and the FlexTail device to represent environment-integrated vs. body-worn HAR systems. In particular, we

develop a recording protocol for selected ADLs,provide a synchronized ADL dataset for FlexTail and camera data,determine the optimal window size for time series classification (TSC),compare TSC classifiers for FlexTail and camera data,evaluate the feasibility of the FlexTail device for HAR anddetermine the system’s performance using $F_{1}$ scores.

## 2. Methodology

### 2.1. Activities

There is no common agreement on the type of activities that are relevant for HAR in medical applications. Minh Dang et al. [8] reported an average of 138 and 8 activities for vision-based and sensor-based datasets, respectively. We compile notable works regarding activities, sensors used to record the data, number of participants recorded, and duration of recording but limit our compilation to articles with less than 20 activities (Table 1).

Based on this review, we select the most frequent indoor (which excludes running and cycling) and in-room (which excludes upstairs and downstairs) activities (Table 2) and augment them with activities to cover a broader spectrum of spine activation. In particular, we include loading and unloading the dishwasher because these actions involve bending, straightening, and trunk torsion. We also include food cutting and eating to determine if FlexTail data can differ activities with slight spine movements.

This yields a total of eleven activities: sitting down, standing up, sitting, standing, walking, loading the dishwasher, unloading the dishwasher, wiping the table, vacuuming, eating, and handling food (cutting). We group the selected activities into a hierarchy of (Figure 1):Mobility (static, transition, dynamic),Housework (cleaning, handling dishwashing), andFood (cutting, eating)

**Figure 1 sensors-25-03806-f001:**
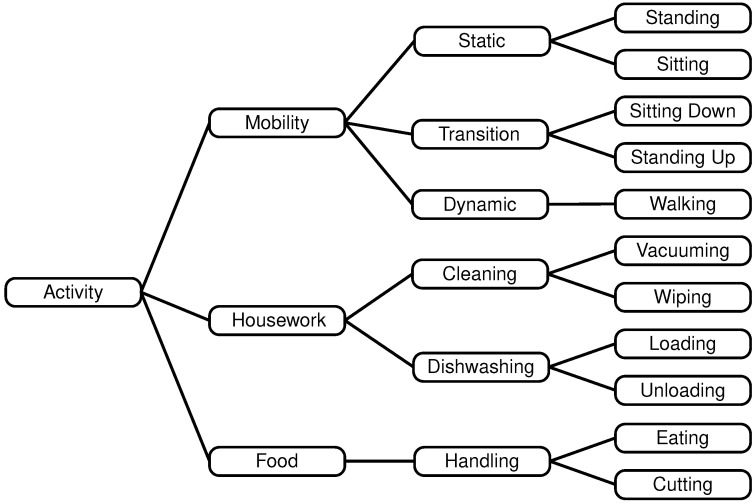
Hierarchy with three, six, and eleven activities.

### 2.2. Protocol

Each participant performs every activity for 1 min, except for “Mobility” activities. Here, the participant sits on a chair to start. When the experiment conductor gives the start signal, the participant stands up and walks to a marked spot on the ground (at a distance of 7 m from the chair, because the average walking speed of 4.5 km/h [29] translates to 5.6 s of walking time, roughly matching the durations of the sitting and standing activities). After standing for 5 s, we instruct the participant to walk back to the chair, sit down again, and stay seated for 5 s. We repeat this procedure for 3 min. The duration of 3 min provides approximately 1 min each of walking, sitting and standing. In total, the recording length for each participant is 10 min. Furthermore, we set the number of participants to ten, which results in a total of 100 min of recordings.

For housework activities, the participants vacuum within a 1.2 m × 1.2 m square marked on the ground, and we allow them to move outside the square while vacuuming. They also wipe a table, which stands at 75 cm in height, with a wiping cloth. Furthermore, they continue to load an already preloaded dishwasher with plates, mugs, glasses, and cutlery. This ensures sufficient material for the unloading phase performed by the participants again.

For the “Handling Food” activities, participants cut a cucumber with a knife and eat the slices while sitting.

### 2.3. Data Collection

As we directly compare FlexTail and video data, we record both simultaneously using the Rectify Studio app (MinkTec GmbH, Braunschweig, Germany). This software records a video with a smartphone camera and transmits the FlexTail data to the recording smartphone via Bluetooth Low Energy (BLE). We set the number of participants to ten.

#### 2.3.1. Video

We record the video with the smartphone’s wide-angle camera (iPhone 13 mini, Apple, Cupertino, CA, USA). We mount the phone on a tripod (C6i Carbon, Rollei, Braunschweig, Germany) with adjustable height. We record the video in 1080p resolution, and we limit the frame rate to 15 fps due to the technical limitations of Rectify Studio. To avoid bias due to fixed views of the activities, we placed the smartphone camera for all activities somewhere in an area with a distance between 1.5 m and 3 m from the participant (Figure 2). We move the tripod between participants to ensure the resulting model does not rely heavily on context, such as recording angle and distance.

#### 2.3.2. FlexTail

The FlexTail device comes in two sizes: 36 cm or 45 cm in length and 2.5 cm in width. It comprises a flexible strain gauge strip and an integrated electronics module (Figure 3). The strain gauge strip, which carries 18 pairs of sensors, enables precise measurement of three-dimensional segment angles at 1° resolution per segment. We facilitate data acquisition at adjustable frequencies ranging from 1 to 100 Hz, allowing capture of both slow and rapid spinal movements. The electronics module, which is housed in a compact plastic casing measuring 4 cm × 4.5 cm, incorporates a 3-axis accelerometer, 3-axis gyroscope, a lithium-ion battery, internal memory, a microprocessor, and wireless connectivity via BLE and USB-C. The FlexTail comes with tight-fitting shirts in five sizes: XS, S, M, L and XL (Figure 3). A longitudinal pocket holds the sensor onto the spine of the subject and prevents slipping or axial rotation.

### 2.4. Data Processing

The code for data processing and training appears on GitHub (https://github.com/onecalfman/paper-flextail-vs-camera.git), accessed on 17 June 2025.

#### 2.4.1. Labeling

We label the activities using Rectify Studio. The app visualizes the video and the FlexTail data side by side. We annotate the current activity by accordingly labeling their start and end points.

We define the start of the standing up activity when the participant begins to stand up. The standing up activity ends when the participant takes a step. This concept is likewise applied for sitting down. We exclude the turning phases after standing from the data but we regard minor movements to adjust the sitting position as part of the sitting activity.

#### 2.4.2. Video

We process the video using the library OpenPose 1.7.0 [30]. This model outputs 25 keypoints per frame that represent the human body’s joint coordinates and a confidence rating, which we store as a JavaScript object notation (JSON) file. OpenPose is a well-established library with predictable computational performance. While newer methods may achieve higher benchmark accuracy, OpenPose provides mature pretrained models without requiring optimization, making it an appropriate foundation for our ADL recognition [31].

#### 2.4.3. FlexTail

We export the FlexTail data from Rectify Studio. The records contain the annotation labels, bending angles, accelerometer and gyroscope data, and a timestamp in milliseconds that is synchronized with the frames from the video.

### 2.5. Classifier

There are special classifiers to measure similarity in time series. Early studies use distance-based methods, such as dynamic time warping. Newer classifiers, such as the exceptionally fast and accurate time series classification using random convolutional kernels (ROCKET) and the random dilated shapelet transform (RDST), address scalability issues in the classification of time series data [32]. Middlehurst et al. [33] present a comprehensive benchmark and provide a library with implementations of 29 algorithms [34], and they identify a group of eight top performing classifiers. However, two of them are not suitable for multivariate time series. Thus, we train six classifiers, QUANT, RDST, FreshPRINCE, InceptionTime, MultiRocketHydra and HIVE-COTE 2.

### 2.6. Training

We train this six classifiers on a workstation with a CPU (Intel i9-14900K, Intel, Santa Clara, CA, USA) and a GPU (A4500, Nvidia, Santa Clara, CA, USA) using the library Aeon 1.0.0 [34] with the default parameters.

We generate a training report and a confusion matrix for all classifiers using scikit-learn [35]. With respect to targeting on real-time tracking, we choose window sizes w = {1,2,3,4} s. We compare results based on the mean $F_{1}$ score over all activities $\bar{F_{1}}$ for each classifier. We calculate the $F_{1}$ score from the precision *P* and recall *R* of the classifier:(1)$$F_{1}=2\frac{PR}{P+R}\;\mathrm{and}\;\bar{F_{1}}=\frac{1}{N}\sum_{i=1}^{N}F_{1,i}$$
where N=11 is the number of activities, and $F_{1,i}$ is the $F_{1}$ score for the *i*-th activity.

### 2.7. Activity Grouping

We select our activities hierarchically, so we evaluate the performance on grouped activities. We train the random dilated shapelet transform (RDST) for FlexTail data and the QUANT for video data with grouped activities on half of the data for training and testing. We also train a general model with the three main activity classes.

## 3. Results

We recruited 10 healthy subjects with a mean weight of 80.9 kg ± 18.2 kg and a mean height of 177 cm ± 11.7 cm. The group comprised two females and eight males. One participant was left-handed, the others were right-handed. We obtained consent from all participants to partake in this study after informing them about the protocol and sensors used.

As the HIVE-COTE 2 classifier encounters a division by zero error when executing, we excluded it from the further analysis.

### 3.1. Classifiers and Window Sizes

For FlexTail data, RDST performs best ($0.88\leq \bar{F_{1}}\leq 0.91$) across all *w* (Table 3).

For video-based pose data, QUANT performs best ($0.90\leq \bar{F_{1}}\leq 0.93$) for all *w* (Table 4).

The QUANT classifier also performs well for FlexTail data ($0.82\leq \bar{F_{1}}\leq 0.86$). Other classifiers such as the MultiRocketHydra also perform well on both datasets: $0.83\leq \bar{F_{1}}\leq 0.87$ and $0.88\leq \bar{F_{1}}\leq 0.91$ for FlexTail and video-based pose data, respectively.

Classification performance remained consistent across window sizes (Table 3 and Table 4), so we selected w=1 s to enable near real-time processing while maintaining accuracy. This shorter window reduces computational load and supports potential real-time clinical applications. There is no significant difference in the window size, so we use w=1 s, allowing for near real-time activity classifications.

### 3.2. Confusion Matrices

In the following, we use the accuracy as the performance measure.

#### 3.2.1. FlexTail

The FlexTail-based HAR achieves perfect classification for eating and cutting, 97% for wiping, and 94% for vacuuming (Figure 4). We train the RDST classifier with a 1 s window size for the individual FlexTail sensors (IMU and flexible strip), and the accuracy is 71% and 84%, respectively.

Notably, FlexTail fails to differentiate sitting and sitting down as well as loading the dishwasher and unloading the dishwasher, with correct classification in only 89%, 71%, 71% and 76% of the cases, respectively (Figure 4).

#### 3.2.2. Video

The video-based HAR classifies eating at 99% and reaches 100% for cutting, 97% for wiping, and 95% for vacuuming (Figure 5).

Here, all mobility activities yield correct classifications below 90%, but handling the dishwasher achieves 97% for both loading and unloading (Figure 5).

#### 3.2.3. Activity Grouping

The FlexTail-based HAR distinguishes all six main activities above 85%, with transition the lowest at 85% and handling food highest with 100% (Figure 6).

The video-based HAR classifies static and dynamic activities with only 81% and 83% correctness, respectively, while all others are above 90% (Figure 7). It best classifies cleaning, dishwashing and handling food with 97%, 97%, and 100% accuracy, respectively.

The FlexTail data yields accuracies of 99%, 99%, and 96% for mobility, housework, and food, respectively (Figure 8). There is no confusion between food and housework.

On the top-level three activities, video-based HAR achieves 97%, 99%, and 100% for mobility, housework, and food (Figure 9).

## 4. Discussion

In this work, we present a hierarchy of ADL with three, six, and eleven activities and a study protocol to record the synchronized data of two sensor systems: FlexTail as body-worn and video-based pose estimation and an environment-integrated HAR. We tested relevant time series classifiers and determined a feasible time window for nearly real time HAR.

With respect to the rather small number of subjects, the performance differences between both approaches are insignificant. On the detailed hierarchy with eleven activities, video-based HAR outperforms FlexTail, which mixes up loading and unloading of the dishwasher as well as sitting and sitting down. However, it outperforms the video-based approach, distinguishing six and three grouped activities.

Our protocol mainly yielded expected results. We expected the difficulties that FlexTail faced in differentiating eating and cutting, but we underestimated the posture differences of these activities. We observed that the participants leaned over the cucumber they cut but sat more upright during eating, which is likely the reason for the better results than initially expected. These activities were included to test out the limitations of FlexTail-based HAR but did not fulfil this expectation. The dishwashing activities differed from real-world conditions, as the subjects loaded the dishwasher from a cupboard and not from a countertop.

An improved protocol should include more variations of sitting activities to provide a more challenging dataset for FlexTail. When the activities were performed by different individuals, it transpired that the transitions (sitting down, standing up) took longer than expected. Moreover, the walking speed of the individuals was slower than expected. This led to reduced measurements of sitting and standing (37 s in average) and more time for walking (68 s).

We noted differences in individual activity performance. For instance, FlexTail-based HAR recognized standing up and sitting with 95% accuracy, while the video-based HAR reached only 76% and 86%, respectively. We attribute these differences to greater variation in visual data for transitions. While the FlexTail remained fixed for each subject, the camera captured data from different views, yielding more variability in the data.

Sitting is frequently misclassified as standing up, sitting down, and walking, as video struggles to capture the slight movements. The video may emphasize the legs when people sit. Most subjects sit with both legs on the ground, but some cross one leg over the other. The video captured this variation, whereas the FlexTail did not.

However, FlexTail data struggled to differentiate between static and transition activities. We attribute this to label placement. Since we recorded activities as a continuous stream and set labels later, the end of standing up was similar to the beginning of walking. We consider this primarily a feature of the dataset rather than a shortcoming of the classifier.

Combining activities into categories yields less improvement than expected. When we use all data, $\bar{F_{1}}$ improves slightly from 0.92 to 0.94 for FlexTail and from 0.90 to 0.93 for pose data. This suggests that classifiers achieve better performance with more data.

Separate models for IMU and the sensor strip demonstrate the superiority of the strip and confirm improvements by signal fusion. This, however, is contrary to our previous experiments, where we reported accuracies of 80% and 50% for IMU and the strip only, respectively [19]. This is due to the different activities in both. In the work of Haghi et al., more transition movements (such as turns from laying on the side to laying on the back) were featured, which certainly affect the IMU rather than the strip.

Due to the methodological differences and the continuous emergence of new classification models, directly comparing results across studies remains challenging.

Our models achieve an accuracy of 88% and 93% for body-worn and environment-integrated data. Other studies report varying results on their respective datasets (Table 5). For instance, we achieve an accuracy of 97% with prolonged activities [19], but the performance decreases to 76% when we include transition. Furthermore, the activities are more periodic, and we carefully hand-craft the models, combining an long short-term memory (LSTM) model with w=2 s and w=10 s with a convolutional neural network (CNN). Marszałek et al. [12] report the lowest accuracy at 33%, utilizing a support vector machine (SVM). We hypothesize that this results from less sophisticated preprocessing and the diversity in their data based on movie pictures. Altini et al. [21] achieve an accuracy of 94%, also using an SVM. They define broader categories and receive an accuracy of 99% for walking. Tao et al. [13] likewise employ an SVM, achieving accuracies of 72% and 57% on their video-based and IMU datasets, respectively. Again, environment-integrated systems outperform body-worn approaches for large numbers of activities. Rezaei et al. [23] test multiple models, reaching the highest accuracy of 97% with a deep learning (DL) model. They use only five activities that differ in the subject’s posture and the room’s location, which is the reason for the best total result. Sen et al. [14] use radar for sensing and achieve a 97% accuracy by applying t-distributed stochastic neighbor embedding (t-SNE) for clustering combined with a CNN for classification. This work seems the best approach for environment-integrated sensors for HAR. Nishida et al. [24] report an accuracy of 77% with a Gaussian mixture model (GMM) that combines video, audio, and IMU data. They use a similar setup as Tao et al. [13] and presumably face the same challenges.

These findings highlight the significant diversity of sensor modalities, classification methods, protocols, and activities used across different studies in the literature. We conclude that our approach lacks the refinement of some other models but has strong potential for further improvements because we use standard classifiers without tuning hyperparameters.

Another limitation results from the lack of data optimization. Neither the FlexTail data nor the video-based pose data was optimized. In particular, the video-based pose data might benefit from appropriate preprocessing, such as normalizing to a reference frame.

It is likely that the video-based HAR improves by normalizing the pose data to canonical or inertial reference frames. The best performing model (QUANT) at least uses differences of the first and second order. However, the impact of the reference frame needs systematical investigation in future research.

Due to the small sample size, we do not statistically test for significance of accuracies or $F_{1}$ scores. This is planned for future research with a larger group of individuals.

In future work, we focus on hyperparameter optimization and explore personalization via transfer learning to improve accuracy in the real world.

## 5. Conclusions

FlexTail (body-worn) and video-based (environment-integrated) HAR perform equally well.FlexTail captures subtle spinal movements and supports distinguishing between static and transition activities.Video-based HAR outperforms FlexTail on activities that involve arm and hand movements.The RDST classifier delivers optimal performance for FlexTail data, while the QUANT classifier is most effective for camera data.Hierarchical activity grouping does not improve classification performance.The 1 s window enables near real-time classification.

## Figures and Tables

**Figure 2 sensors-25-03806-f002:**
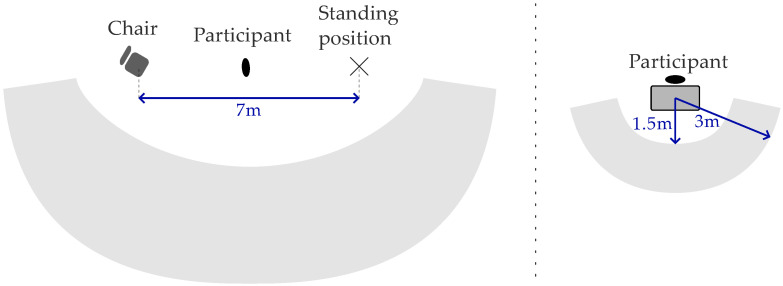
Top view of the recording setup. Left: scenario for the movement activity. Right: scenario for all other activities. The gray rectangle on the right represents the table, dishwasher, or area that is to be vacuumed. The irregular gray shapes are the areas in which the tripod with the camera is placed.

**Figure 3 sensors-25-03806-f003:**
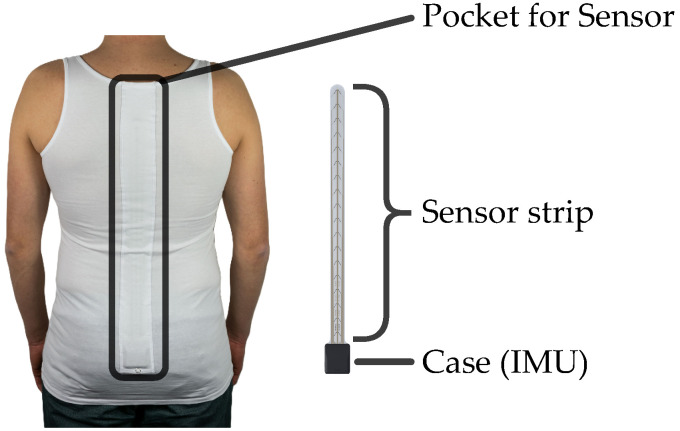
On-body configuration of the FlexTail device. Left: custom shirt with a longitudinal pocket. Right: FlexTail device.

**Figure 4 sensors-25-03806-f004:**
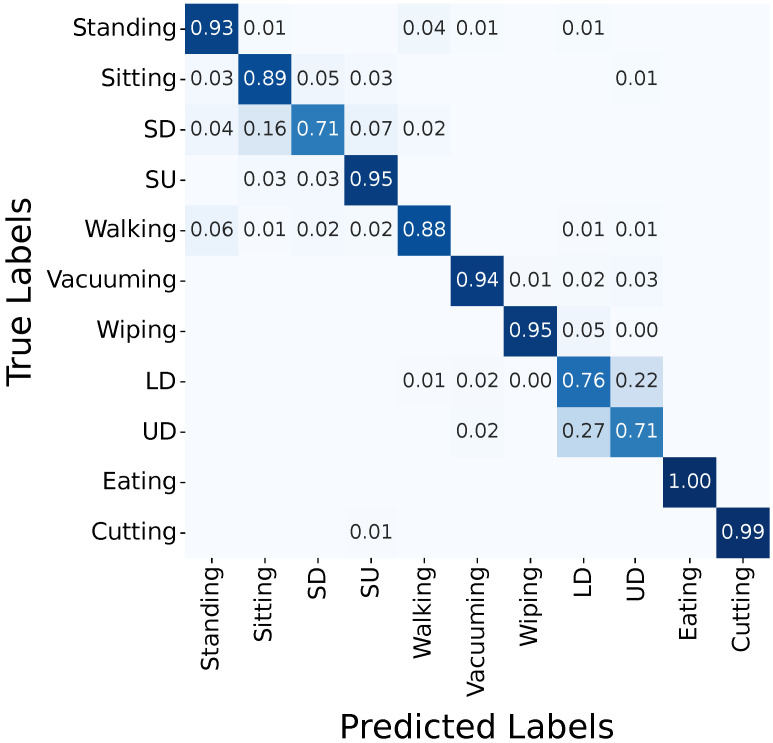
Confusion matrix for FlexTail-based HAR (RDST, w=1 s). Abbreviations: sitting down (SD), standing up (SU), loading dishwasher (LD), and unloading dishwasher (UD).

**Figure 5 sensors-25-03806-f005:**
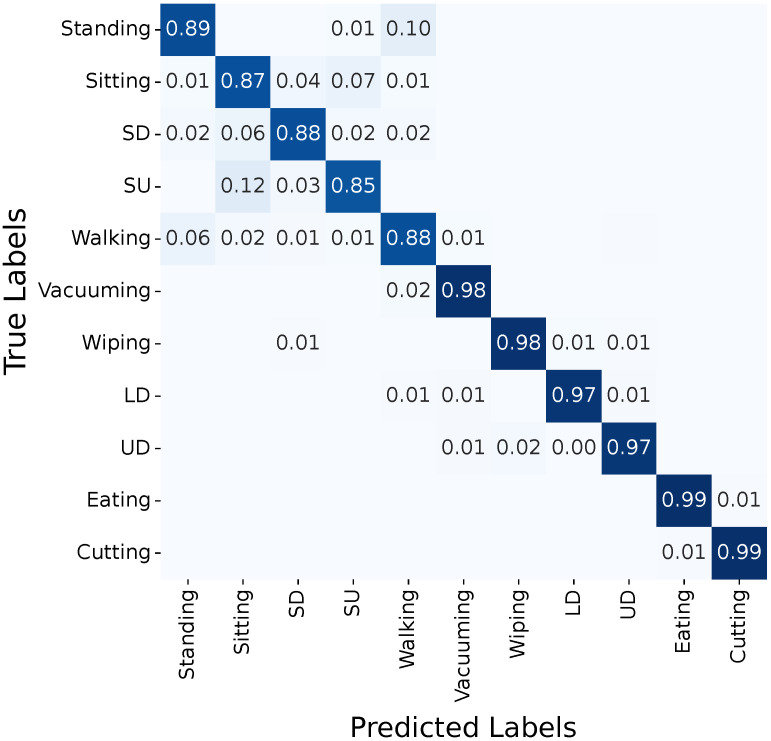
Confusion matrix for video-based HAR (QUANT, w=1 s). Abbreviations: sitting down (SD), standing up (SU), loading dishwasher (LD), unloading dishwasher (UD).

**Figure 6 sensors-25-03806-f006:**
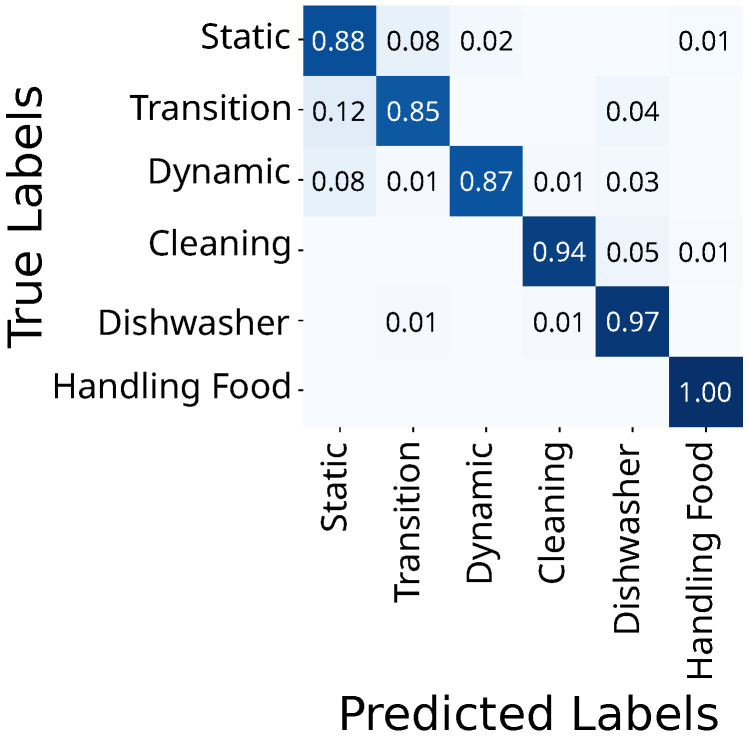
Confusion matrix for FlexTail-based HAR on grouped activities (RDST, w=1 s).

**Figure 7 sensors-25-03806-f007:**
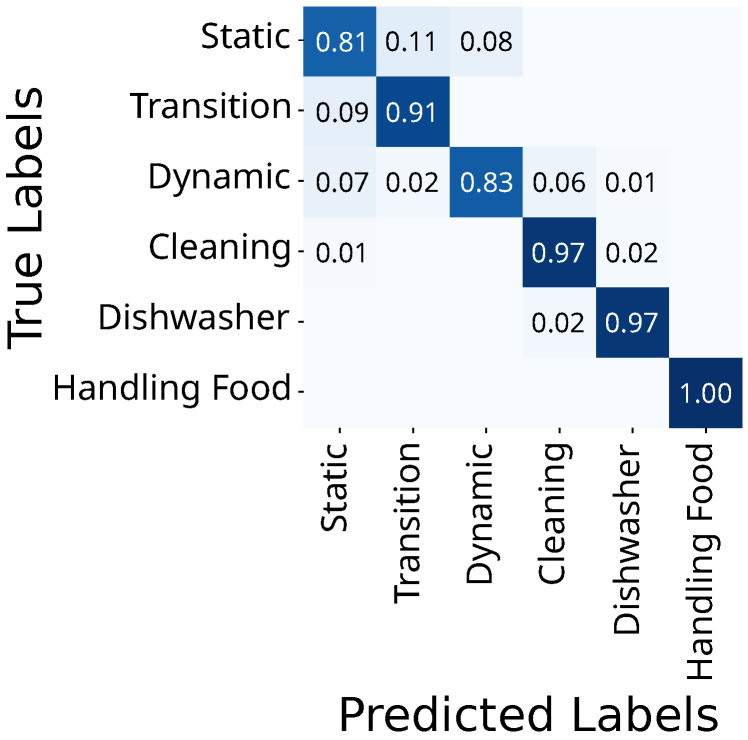
Confusion matrix for video-based HAR on grouped activities (QUANT, w=1 s).

**Figure 8 sensors-25-03806-f008:**
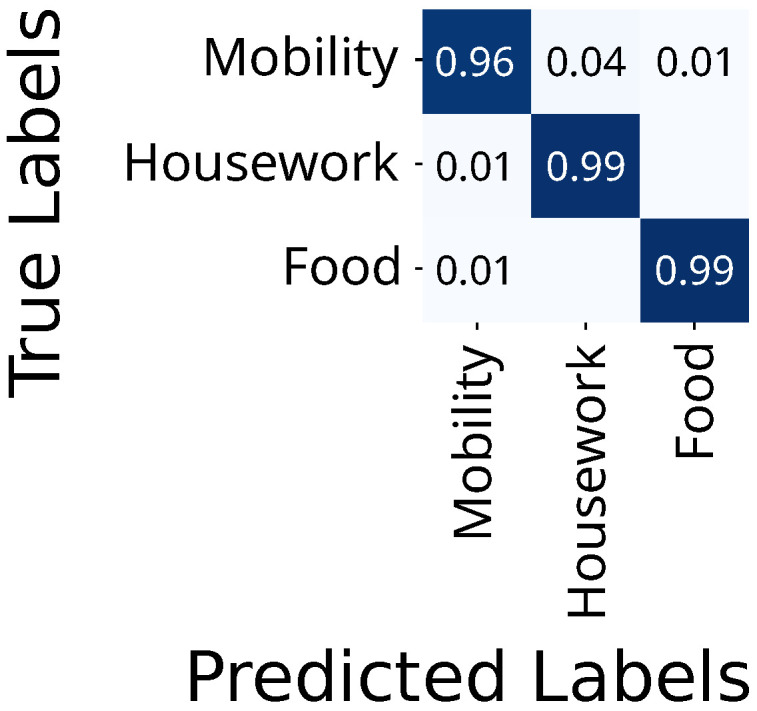
Confusion matrix for FlexTail-based HAR on top-level activities.

**Figure 9 sensors-25-03806-f009:**
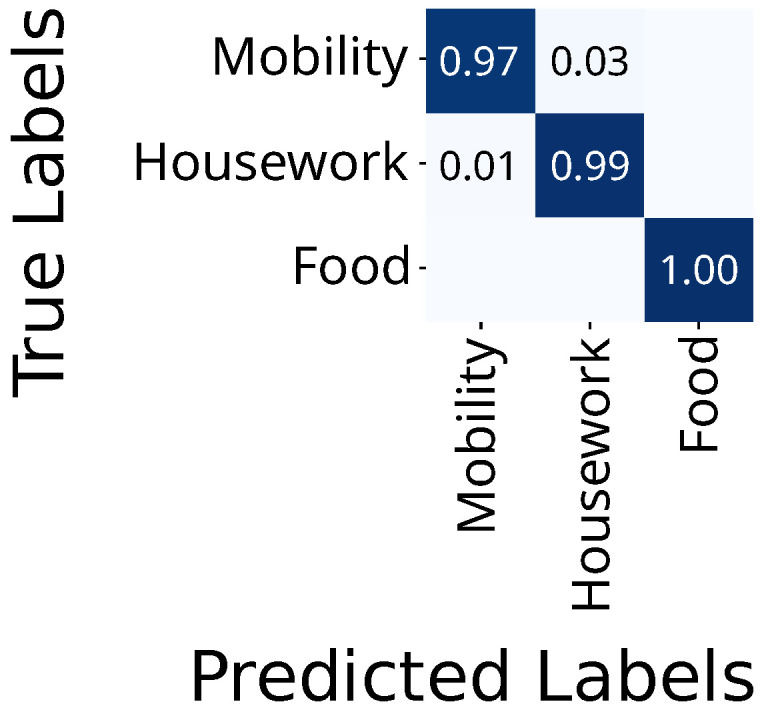
Confusion matrix for video-based HAR on top-level activities.

**Table 1 sensors-25-03806-t001:** Datasets and research papers that serve as a reference for our protocol. The duration is seconds per activity; * indicates the duration over all activities. We include the blue-marked activities in our protocol.

Source	Sensor	Subjects	Duration	Activities
[19]	FlexTail	30	300 *	Walking, Sitting, Standing, Running, Upstairs, Sitting Down, Standing Up, Laying on Side, Laying on Back, Bending Forward, Bending Backward, Bending Right, Bending Left
[12]	Video	-	30	Sitting, Running, Sitting Down, Standing Up, Answer Phone, Driving Car, Eat, Fight Person, Get Out of Car, Handshake, Hug Person, Kissing
[21]	IMU + Heart rate	22	240	Walking, Sitting, Standing, Running, Vacuuming, Sweeping, Cycling, Laying Down, Desk Work, Reading, Writing, PC Work, Stacking Groceries, Washing Dishes, Cleaning
[22]	IMU + Heart rate	9	250 *	Walking, Sitting, Standing, Running, Upstairs, Downstairs, Cycling, Laying, Nordic Walk, Ironing, Vacuuming, Rope Jump, Undefined
[13]	IMU + Depth Camera	5	65 *	Walking, Sitting, Standing, Vacuuming, Sitting Down, Standing Up, Sweeping, Wiping Table, Dusting, Cleaning Stain, Picking Up, Squatting, Stretching
[23]	Low Resolution Infrared Camera	18	60	Walking, Standing, Sitting, Sitting Down, Laying Down, Fall Forward, Falling Backward, Falling Laterally, Falling from Bed
[24]	IMU + Camera	1	6780 *	Cycling, Teeth Brushing, Wiping Table, Preparing Food, Driving Car, Washing, Cleaning Bath, Cleaning Room, Drying Clothes, Bath, Shopping, Toilet, TV, Meal, Notebook, Reading, Office, Smartphone
[25]	IMU	51	180	Walking, Standing, Sitting, Running, Upstairs, Downstairs, Teeth Brushing, Eating (Soup, Chips, Pasta, Sandwich), Typing, Drinking from Cup, Kicking (Soccer Ball), Playing Catch with Tennis Ball, Dribbling (Basketball), Writing, Clapping, Folding Clothes
[26]	IMU	10	200	Walking, Standing, Sitting, Running, Upstairs, Downstairs, Cycling
[14]	Radar	7	173 *	Walking, Sitting, Running, Vacuuming, Eating, Squats, Lunges, Jumping, Clapping, Changing-Clothes, Folding Clothes, Waving, Phone Typing, Phone Talking, Playing-Guitar, Combing-Hair, Brushing, Drinking, Laptop Typing
[15]	WiFi	1	940 *	Walking, Standing, Sitting, Sitting Down, Standing Up, Laying, Absence
[16]	IMU	11	120	Walking, Running, Vacuuming, Cycling, Eating, Wiping Table, Laying Down, Preparing Food, Socializing, Reading, Getting Dressed, Sit to Stand, Laying Down to Stand
[27]	IMU	8	120	Walking, Upstairs, Downstairs, Vacuuming, Sweeping, Teeth Brushing, Eating, Walking Fast, Laptop, Keyboard, Writing, Handwashing, Face Washing, Dusting, Rubbing, Fast
[28]	IMU	9	560 *	Walking, Sitting, Standing, Upstairs, Downstairs, Cycling, Undefined

**Table 2 sensors-25-03806-t002:** Activities that occur more than once in relevant articles (Table 1). We include the blue-marked activities in our protocol.

Activity	Count
Walking	12
Sitting	11
Standing	9
Running	8
Upstairs	6
Cycling	6
Vacuuming	6
Downstairs	5
Sitting Down	5
Standing Up	4
Eating	3
Wiping Table	3
Laying Down	3
Sweeping	3
Teeth Brushing	3
Reading	3
Writing	3
Preparing Food	2
Driving Car	2
Laying	2
Dusting	2
Folding Clothes	2
Clapping	2

**Table 3 sensors-25-03806-t003:** $\bar{F_{1}}$ for FlexTail data.

	Window Size *w* [s]			
Classifier	1	2	3	4
FreshPRINCE	0.84	0.84	0.82	0.85
InceptionTime	0.56	0.44	0.43	0.37
MultiRocketHydra	0.87	0.87	0.86	0.86
QUANT	0.86	0.86	0.82	0.85
RDST	**0.90**	**0.88**	**0.88**	**0.91**

**Table 4 sensors-25-03806-t004:** $\bar{F_{1}}$ for video-based pose data.

	Window Size *w* [s]			
Classifier	1	2	3	4
FreshPRINCE	0.87	0.87	0.87	0.85
InceptionTime	0.82	0.84	0.76	0.70
MultiRocketHydra	0.88	0.89	0.91	0.89
QUANT	**0.90**	**0.91**	**0.93**	**0.90**
RDST	0.86	0.88	0.91	**0.90**

**Table 5 sensors-25-03806-t005:** Reported accuracies. If authors group activities, their number differs from Table 1. Abbreviations: body-worn (BW), environment-integrated (EI), long short-term memory (LSTM), deep learning (DL), Gaussian mixture model (GMM).

Source	Year	Model	BW/EI	No. Activities	Total	Standing	Sitting	Sitting Down	Standing Up	Walking	Vacuuming	Wiping	Loading	Unloading	Eating	Cutting
Ours	2025	RDST	BW	11	0.88	0.93	0.89	0.71	**0.95**	0.88	0.94	0.95	0.76	0.71	**1.00**	**0.99**
	2025	QUANT	EI	11	0.93	0.89	0.87	**0.88**	0.85	0.88	**0.98**	**0.98**	**0.97**	**0.97**	0.99	**0.99**
[19]	2023	CNN	BW	12	0.94	**0.97**	0.87	-	-	0.91	-	-	-	-	-	-
	2023	LSTM	BW	12	0.96	0.76	0.83	-	-	0.92	-	-	-	-	-	-
	2023	CNN + LSTM	BW	12	**0.97**	0.78	**0.91**	-	-	0.96	-	-	-	-	-	-
[12]	2009	SVM	EI	12	0.32	-	-	0.32	0.35	-	-	-	-	-	0.29	-
[21]	2016	SVM	BW	6	0.94	-	-	-	-	**0.99**	-	-	-	-	-	-
[13]	2015	SVM	BW	13	0.57	0.80	0.63	0.35	0.44	0.77	0.62	0.66	-	-	-	-
	2015	SVM	EI	13	0.72	0.77	0.77	0.84	0.86	0.86	0.75	0.67	-	-	-	-
	2015	SVM	BW + EI	13	0.75	-	-	-	-	-	-	-	-	-	-	-
[23]	2023	DL	EI	5	**0.97**	-	-	-	-	-	-	-	-	-	-	-
[24]	2014	GMM	EI + BW	13	0.77	-	-	-	-	-	-	-	-	-	-	-
[14]	2024	t-SNE + CNN	EI	9	**0.97**	-	-	-	-	-	-	-	-	-	-	-
[16]	2011	KNN	BW	13	-	-	-	-	-	-	-	-	-	-	-	-
[27]	2021	SVM	BW	17	0.91	-	-	-	-	0.87	0.87	-	-	-	0.97	-

## Data Availability

Due to GDPR restrictions, the data used in this study cannot be shared publicly.

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
