# Peer review of "Wearable Spine Tracker vs. Video-Based Pose Estimation for Human Activity Recognition"

_sensors, 2025, doi:10.3390/s25123806_

Round 1
Reviewer 1 Report
Comments and Suggestions for Authors
Dear authors,
Thank you for your concise paper. Below are my detailed comments and suggestions for improvement:
Minor Corrections and Typos
- Line 32: A punctuation mark appears to be missing at the end of the sentence.
- Line 34: The phrase “in the category” is unclear—please clarify which category you are referring to.
- Line 65: Consider removing the superfluous "a"
- Line 127: Typo in “which we store aS”.
- Line 184: Consider using “distinguishes” rather than “differs” to maintain grammatical consistency.
- Line 195: Typo in “FleXTail”.
- Line 206: Do you mean "countertop" instead of "cutlery"? As written, the sentence is unclear.
Suggestions for Clarification and Improvement
- FlexTail Illustration: Including a visual representation of the FlexTail device and particularly the shirt/pocket system would enhance reader understanding. While such images may be available in [14], reproducing them in this paper would be helpful. Additionally, clarifying how tightly the shirt fits may be relevant for interpreting sensor stability.
- Sensor Contribution: Have you analyzed the relative importance of the different FlexTail sensing modalities (flex sensors vs. accelerometer/gyroscope)? For instance, in activities like "cutting," it seems plausible that inertial data is more informative than flexion. A brief discussion or ablation study would strengthen your methodological transparency.
- 2D Pose Normalization: Although you vary the camera position to reduce reliance on context (lines 98 and 210), OpenPose outputs 2D, camera-relative features. Consider normalizing these data to a canonical or inertial reference frame—this could yield more robust, subject-independent models and improve comparison with FlexTail.
- "Posture Differences" (Line 204): Referring to these as "posture differences" might suggest static variation. However, the performance gap may be better explained by dynamic differences that are more visible in inertial sensor data than in flexion measurements.
- Since you are investigating the performance of your own product, greater care seems necessary to convince the reader that equal optimization work was put into the camera-based methods.
- It seems that the activity selection favors "spine heavy" activities. It is clear that activities with more use of extremeties would perform better in video. This should be discussed.
- Tests for significance of F1 scores are missing. With such a small sample set, are they significant?
Additional Reference
For Section 2.5, I encourage you to consider citing the following work:
Khandelwal et al., Posture Classification based on a Spine Shape Monitoring System, in ICCSA 2019 (DOI: [10.1007/978-3-030-24311-1_36])
This study demonstrates the effectiveness of traditional machine learning techniques (Extra Trees, AdaBoost, ANN) for wearable spine tracking and explores both inter-subject and personalized classification—topics closely related to your methodology and discussion. Furthermore it shows a wearable system that monitors spine shape in a similar way (without flex sensor, only acc/gyr).
Thank you again for your contribution. I believe that with these revisions, your work will offer greater clarity and impact.
Best regards
Reviewer 2 Report
Comments and Suggestions for Authors
This paper presents a comparative study with some contribution to HAR research. However, the novelty of the research is not well presented. Need to compare the proposed research with more studies to show the novelty.
Reviewer 3 Report
Comments and Suggestions for Authors
The paper presents a comparison between a wearable (FlexTail) and a video-based (OpenPose) system for human activity recognition. The topic is relevant, and the overall structure of the paper is clear, but several aspects need clarification or improvement.
- Introduction
- The motivations are weak. What real-world problem or clinical need does this comparison address? The claim that "most papers on HAR consider only a single type of input" is not sufficient justification for the study.
- It is unclear why exactly the two technologies (FlexTail and OpenPose) were selected. Are these the most promising or accessible? Why not compare with other technologies?
- There is no evidence that this work was derived from actual user or clinical requirements.
- Methodology
- In Section 2.1, the choice of activities is not well justified. Why these 11? Why not use a standardised set from commonly used datasets for comparison?
- The protocol description in Section 2.2 should be better explained.
Is "eating the slices" done sitting or standing?
"3 min mobility procedure": how was this duration decided?
- In Section 2.3.1 visual reference (e.g., photo or schematic) should be provided for the video setup. Without such a reference, reproducibility is difficult. Where is the smartphone placed in the room?
- In Section 2.6, the sentence “We train this six classifiers” should be corrected for grammar. Moreover, the list of the six classifiers should be explicitly stated in the main text and not inferred from tables.
- Results
- There is no detailed discussion on why the HIVE-COTES2 classifier failed. What were the errors or computational issues?
- The authors claim that activities are classified “in nearly real-time” with 1-second windows. However, real-time feasibility is not experimentally validated. Was it measured?
- In Table 5, the publication years should be added: lower-performing results seem related to older studies (>=10 years old).
- and 5. Discussion and Conclusion
- The discussion is generally well-structured but lacks critical insight. Some findings are simply restated.
- The final section (Conclusions) is a bullet-point list, which is stylistically poor and insufficiently analytical for a scientific journal. Conclusions should synthesise insights, limitations, and future directions in a narrative form.
Other comments
- Table and figure references should be introduced and discussed in the text before the reader encounters them.
- Line 34: After “category”, probably a word is missing.
- Section 2.2, first sentence: I would refer to “Each subject” instead of “All subjects”
